# The Kinetics of the Redox Reaction of Platinum(IV) Ions with Ascorbic Acid in the Presence of Oxygen

**DOI:** 10.3390/ma16134630

**Published:** 2023-06-27

**Authors:** Magdalena Luty-Błocho, Aleksandra Szot, Volker Hessel, Krzysztof Fitzner

**Affiliations:** 1Faculty of Non-Ferrous Metals, AGH University of Krakow, al. Adama Mickiewicza 30, 30-059 Kraków, Poland; szot@agh.edu.pl (A.S.);; 2School of Chemical Engineering, The University of Adelaide, Adelaide 5005, Australia; volker.hessel@adelaide.edu.au

**Keywords:** redox reaction, Pt(IV) ions, kinetic rate constant, photolysis, stopped-flow spectrophotometry

## Abstract

In this work, the kinetics of the redox reaction between platinum(IV) chloride complex ions and ascorbic acid is studied. The reduction process of Pt(IV) to Pt(II) ions was carried out at different reagent concentrations and environmental conditions, i.e., pH (2.2–5.1), temperature (20–40 °C), ionic strength (I = 0.00–0.40 M) and concentrations of chloride ions (0.00–0.40 M). The kinetic traces during the reduction process were registered using stopped-flow spectrophotometry. Based on the kinetic traces, the rate constants were determined, and the kinetic equations were proposed. It was shown that in the mild acidic medium (pH = 2.5), the reduction process of Pt(IV) to Pt(II) ions is more complex in the presence of oxygen dissolved in the aqueous solutions. For these processes, the values of the enthalpy and entropy of activation were determined. Moreover, the mechanism of the reduction of Pt(IV) to Pt(II) ions was proposed. The presented results give an overview of the process of the synthesis of platinum nanoparticles in the solution containing oxygen, in which the reduction process of Pt(IV) to Pt(II) ions is the first step.

## 1. Introduction

The reduction process of the Pt(IV) ions was studied in detail by Elding et al. [1], Wojnicki et al. [2] and Jovanović et al. [3]. They showed that the reduction process of Pt(IV) to Pt(II) ions is usually pseudo-first-order, and the reduction process of metal ions depends strongly on the pH of the solution, which is related to the form of the metal precursor or reducing agent. However, to prevent Pt(II) ions from undergoing secondary oxidation to Pt(IV) ions, the experiments were mostly carried out under specific environmental conditions, i.e., in deaerated solutions with the presence of sodium chloroacetate, phosphate buffers and chelate ions like EDTA [1]. These previous investigations are the gate to a better understanding of reaction kinetics, which takes place in the system containing platinum ions. Despite the fact that the kinetics and mechanism of Pt(IV) ion reduction are well described in the literature, especially by Elding’s group [4,5,6,7], these data are difficult to use if there is a need to synthesize the nanomaterials. Usually, the process of platinum nanoparticle formation contains at least two stages, in which the reduction process of Pt(IV) to Pt(II) ions is the first step. The next one is related to the nucleation and autocatalytic particle growth that was described by the model of Watzky–Finke (W–F) [8]. This model was successfully applied by other researchers [9,10,11,12]. However, there are still many questions that remain without answers. What is especially interesting is the role of oxygen dissolved in the aqueous solution and its influence on the processes taking place during the platinum nanoparticle formation step by step.

In this context, it became interesting to investigate how the presence of oxygen influences kinetics and the mechanism of Pt(IV) ion reduction using ascorbic acid. In our previous work about Pt(IV) ion reduction using sodium borohydride, we showed that the presence of oxygen changed the reaction mechanism [13]. Thus, we decided to examine the implications related to the presence of oxygen in the reacting system when ascorbic acid is used as the reducing agent. The choice of this compound is also related to its popularity in the chemical synthesis. It is often used as a reducing agent of copper, iron [14], gold [15] and platinum [7] ions. Also, due to its mild reducing properties as compared to sodium borohydride, it is applied for the synthesis of silver [16], gold [17], palladium [9] and platinum [18] using a chemical method. The chemistry of this compound, due to its biological role in living organisms, is very complex. It oxidizes to different forms, which depend mainly on environmental factors like pH, standard reduction potentials and oxygen content. The process of ascorbic acid oxidation was described by Tu et al. [19] in detail. The reactivity of the ascorbic acid may have many consequences in the course of the reaction. Hence, the presence of oxygen in the solution may highly affect the process of ascorbic acid oxidation and the Pt(IV) to Pt(II) ion reduction mechanism. Consequently, it has an influence on the final platinum nanoparticles.

## 2. Materials and Methods

### 2.1. Reagents

Chloride platinum (IV) complex. The stock solution of platinum(IV) ions was 0.0763 M, and solution of [PtCl_6_]^2−^ was prepared according to reference [20]. The pH of the solution was fixed with the addition of HCl (p.a., POCH, Poland) and NaOH (p.a. POCH, Poland). The concentration of chloride ions was determined using NaCl (p.a., POCH). Ionic strength was maintained using NaClO_4_ (p.a., Koch-Light Laboratories Ltd., Suffolk, UK).

L-ascorbic acid (H_2_Asc). The aqueous solution of the reducer was prepared by dissolving the appropriate weight of L-ascorbic acid (p.a., Sigma-Aldrich, St. Louis, MO, USA) in deionized water. Depending on the reaction conditions (pH of the solution), L-ascorbic acid has three different forms including undissociated (H_2_Asc), partially dissociated (HAsc^−^) and completely dissociated (Asc^2−^). Its stability diagrams can be found elsewhere [15,21]. 

### 2.2. Methods of Analysis

To measure the reaction kinetics, a stopped-flow spectrophotometer (Applied Photophysics, Leatherhead, UK) was used, operating in the 190–900 nm wavelength range and enabling the measurement of ms response. The spectra of reagents were registered using UV–Vis (190–900 nm) spectrophotometry (Shimadzu, Kyoto, Japan) equipped in thermostatic cell. An EPS-505 pH meter (Elmetron, Zabrze, Poland) was used to measure the pH of the solutions. 

## 3. Results and Discussion

### 3.1. Experimental Conditions

The kinetics of the reduction reaction of the platinum (IV) chloride complex using L-ascorbic acid was carried out at various excesses of reductant compared to the precursor, initial reagent concentrations, temperature range, ionic strength and chloride ion content. The experimental conditions are summarized in Table 1.

### 3.2. Spectra of Reagents

Spectrophotometric studies have shown that both L-ascorbic acid and the platinum(IV) chloride complex have characteristic spectra. For L-ascorbic acid in an aqueous solution at low concentration values (2 × 10^−4^ M), the characteristic maximum absorbance appears at the wavelength of λ_max_ = 254 nm (Figure 1a, A), whereas for the Pt(IV) ions at concentrations of ~10^−4^ M, the peak is localized at 262 nm (Figure 1a, B). It is worth noting that the appearance of the peak at 262 nm is not limited to the [PtCl_6_]^2−^ form. Murray et al. [22] studied the Pt(IV) ion spectra and reported that species like [PtCl_5_H_2_O]^−^ and [PtCl_6_]^2−^ have analogical spectra, with the minima and maxima located at 230 and 262 nm, respectively. The difference between them are in the molar coefficient values, and for [PtCl_5_H_2_O]^−^;, they are equal to ε_230 nm_ = 12,500 M^−1^cm^−1^ and ε_262 nm_ = 11,600 M^−1^cm^−1^, whereas for [PtCl_6_]^2−^, they are equal to ε_230 nm_ = 3300 M^−1^cm^−1^ and ε_262 nm_ = 24,500 M^−1^cm^−1^ [22]. At higher Pt(IV) ion concentrations, the maxima of both species are located at a similar wavelength, i.e., 353 nm, and have the same value of the molar coefficient (ε_353 nm_ = 490 M^−1^cm^−1^); they are difficult to separate.

Figure 1a confirms that the characteristic spectra derived for the individual reagents overlap at low reagent concentrations. Therefore, to observe the disappearance of the characteristic band of [PtCl_6_]^2−^, the wavelength at 360 nm was selected (Figure 1a) due to its stronger intensity as compared to the peak located at 455 nm (Figure 1a, C).

After mixing the solutions containing platinum(IV) ions and reductant at a volume ratio of 1:1, changes in the characteristic spectrum for the metal ions were observed over time (Figure 1b). The decrease in the absorbance (for wavelength, λ_max_ = 360 nm, ε_360 nm_ = 496.5 M^−1^cm^−1^) from approx. 0.4 to 0.2 (Figure 1b), as well as the “raising” of the spectrum in the detection area (Appendix A), suggests that the reduction reaction between the reactants takes place according to Equation (1) and with a rate constant *k*.
(1)$$\mathrm{Pt}\left(\mathrm{IV}\right)+H_{2}\mathrm{Asc}\overset{k}{\to }\mathrm{products}$$

As a result of this reaction, the platinum(IV) chloride complex is reduced to Pt(II) (Figure 1b) and then to Pt(0), while the ascorbic acid is oxidized. An additional confirmation of metallic particles being obtained in the reduction process was the change to a black color in the tested solution and an increase in the absorbance at the entire wavelength range resulting from the appearance of turbidity in the solution (see Appendix A).

Considering the stability diagram of the Pt(IV) ions [23] and the hydrolysis process (2) as well as the form of the reducing agent in typical experimental conditions, i.e., pH = 2.5 (pH determined after mixing reactants), it can be assumed that the process of reduction involves both forms, i.e., [PtCl_6_]^2−^ and [PtCl_5_H_2_O]^−^, emerging during the following hydrolysis process:(2)$${\left[{\mathrm{PtCl}}_{6}\right]}^{2-}+H_{2}O\overset{K_{1,k_{\mathrm{aq}}}}{↔}\left[{\mathrm{PtCl}}_{5}(H_{2}O\right){]}^{-}+{\mathrm{Cl}}^{-}$$
where K_1_—equilibrium constant, (K_1_ = 0.032). 

By comparing the experimental spectra (Figure 1a, B) and the data from the literature, it can be seen that the value of the molar coefficient determined at 262 nm equals 25,000 M^−1^cm^−1^ (($A=\varepsilon \times l\times C_{Pt\left(IV\right)})$, *l*–path length in cm) and it is close to 24,500 M^−1^cm^−1^ [22]. Thus, under reaction conditions, the peak on the spectrum comes from [PtCl_6_]^2−^. Moreover, the data in [24] shows that the rate of the hydrolysis process (2) is very slow and the value of the rate constant (*k*_aq_) equals ∼5 × 10^–7^ s^–1^ at 25 °C. However, Archibald et al. observed that the reaction of (2) is much faster during daylight exposition [25,26]. They postulated that the conversion of ${\left[{\mathrm{PtCl}}_{6}\right]}^{2-}$ to $\left[{\mathrm{PtCl}}_{5}(H_{2}O\right){]}^{-}$ ions is bound by the effect of the photoaquation of platinum complexes, which was thoroughly investigated by Wickramasinghe et al. [27,28,29]. Taking into account the above aspects, in our reacting system, at the beginning of the process, we have one form of a metal precursor, i.e., ${\left[{\mathrm{PtCl}}_{6}\right]}^{2-}$. However, the share between the two forms [${\mathrm{PtCl}}_{6}{]}^{2-}$ and $\left[{\mathrm{PtCl}}_{5}(H_{2}O\right){]}^{-}$) is dependent on the pH, daylight exposition and UV irradiation [30]. Also, the reductant (ascorbic acid) may exist as H_2_Asc, HAsc^−^ and Asc^2−^ [7,31] depending on the pH of the solution. Thus, the reduction process of Pt(IV) to Pt(II) is more complex. For this reason, in this work, we focus on the first step in the process of platinum nanoparticle formation, i.e., the reduction of Pt(IV) to Pt(II) ions, underling the consequences of the presence of oxygen in the solution and its influence on the reaction path.

### 3.3. Kinetic Curves Determination and Proposed Mechanism 

Based on the obtained experimental spectra evolution (Figure 1b), the kinetic curves, i.e., the absorbance change in time, were registered. Taking into account the fact that the change in the absorbance value at 360 nm is very fast at the beginning, we decided to register the kinetic curves using stopped-flow techniques. This spectrophotometer allows for a fast solution heating and protects the solutions from oxygen from the air (the system is closed compared to standard spectrophotometer equipped with exchange cuvette), and also allows for fast reagent mixing as well as analysis. Next, the values of absorbance were recalculated according to the Lambert–Beer law (A ∝ C) and drawn as a function of time. The sample of the kinetic curve, illustrating the decrease in the platinum concentration over time at 360 nm and fitting the equation to the experimental data, is shown in Figure 2. 

If we look at the course of the kinetic curve, it seems that about 20 s after mixing, the additional process is activated (Figure 2). As a result, the overall rate of the reaction slows down. 

During the reduction of Pt(IV) ions using ascorbic acid, Pt(II) ions are formed, which may undergo secondary oxidation to Pt(IV) according to the suggestion by Elding et al. [7]. The question is, which compound plays the role of the oxidizing agent of Pt(II) ions in the reacting system? Taking into account that ascorbic acid reacts with oxygen in aqueous solutions, we follow the reaction path and analyze which compound might be reactive and oxidize Pt(II) to Pt(IV). 

The reaction path between ascorbic acid and oxygen dissolved in the aqueous solution under pulse radiolysis and certain reaction conditions was described by Cabelli et al. [32]. According to this work, in the oxidation process, the reactive compound is one form, i.e., partially deprotonated ascorbic acid (HAsc^−^), which is formed in the following reaction:(3)$$H_{2}\mathrm{Asc}\overset{H_{2}O,{\;K}_{a}}{↔}{\mathrm{HAsc}}^{-}+H^{+}$$

Then, the partially deprotonated form of ascorbic acid is oxidized with oxygen dissolved in water, and the reaction products are free radicals (4) or dehydroascorbic acid (DHA) and hydrogen peroxide (5), which are known from the literature and described by Cabelli et al. [32] and Jiang et al. [33].
(4)$${\mathrm{HAsc}}^{-}+O_{2}\overset{}{\to }{\mathrm{HAsc}}^{-}+O_{2}{}^{-}+H^{+}$$
(5)$${\mathrm{HAsc}}^{-}+O_{2}+H^{+}\overset{}{\to }\mathrm{DHA}+H_{2}O_{2}$$

The ionic oxygen radical (O_2_^−^) is unstable, and in an acidic environment, it forms the radical HO_2_ [34]. However, these radicals are formed under pulse radiolysis, which are far from our experimental condition. In this context, the findings described by Jiang et al. [33] are more relevant to our studies. They postulated that the oxidation of ascorbic acid via molecular oxygen is facilitated by Cu(II) complexes with amyloid peptides and its aggregates. They also showed that free Cu(II) ions, in the presence of ascorbic acid and copper complexes, facilitate the reduction of oxygen to hydrogen peroxide as a main product of the process [33]. 

The results obtained by Murray et al. [22] are also interesting. They studied the oxidation of Pt(II) to Pt(IV) ions using hydrogen peroxide. They assumed that Pt(II) ions can also be oxidized by other compounds appearing in the reacting solutions, i.e., reactions (6)–(9). Moreover, Pt(IV) ions can be in two forms depending on the reactive compound, like hypochlorous acid and chlorine, as seen in (7) and (9).
(6)$$H_{2}O_{2}+H^{+}+{\mathrm{Cl}}^{-}\to \mathrm{HOCl}+H_{2}O$$
(7)$$\mathrm{HOCl}+{\left[{\mathrm{PtCl}}_{4}\right]}^{2-}+H^{+}\to {\left[{\mathrm{PtCl}}_{5}\left(H_{2}O\right)\right]}^{-}$$
(8)$$\mathrm{HOCl}+H^{+}+{\mathrm{Cl}}^{-}↔{\mathrm{Cl}}_{2}+H_{2}O$$
(9)$${\left[{\mathrm{PtCl}}_{5}\left(H_{2}O\right)\right]}^{-}+{\mathrm{Cl}}_{2}+{\mathrm{Cl}}^{-}+{\left[{\mathrm{PtCl}}_{4}\right]}^{2-}\to 2{\left[{\mathrm{PtCl}}_{6}\right]}^{2-}+H_{2}O$$

In our reacting system, the Pt(IV) ions are reduced to Pt(II), which might also be oxidized using H_2_O_2_ produced during the process of ascorbic acid oxidation (5). 

Taking into account that the process is complex, the rate of the Pt(IV) complex reduction can be described as a sum of two processes appearing at pH = 2.5 as follows:(10)$$\gamma_{\mathrm{Pt}\left(\mathrm{IV}\right)}=\gamma_{\alpha -\mathrm{Pt}\left(\mathrm{IV}\right)}+\gamma_{\beta -\mathrm{Pt}\left(\mathrm{IV}\right)}$$

Hence, the differential equation describing the reduction of Pt(IV) ions using ascorbic acid can be described as a sum of two processes (Equation (10)). The first one relates to the reduction of Pt(IV) ($\alpha -Pt\left(IV\right))$ to Pt(II), and the second one relates to the oxidation of Pt(II) ions to Pt(IV) ($\beta -Pt\left(IV\right))$. Alternatively, both processes relate to Pt(IV) ion reduction, in which two forms shown in (2) coexist. The differential equation has the following form:(11)$$\frac{{\mathrm{dC}}_{\mathrm{Pt}\left(\mathrm{IV}\right)}}{\mathrm{dt}}=\frac{{\mathrm{dC}}_{\alpha -\mathrm{Pt}\left(\mathrm{IV}\right)}}{\mathrm{dt}}+\frac{{\mathrm{dC}}_{\beta -\mathrm{Pt}\left(\mathrm{IV}\right)}}{\mathrm{dt}}$$

In Equation (11), each stage is pseudo-first-order (due to the application of isolation conditions), and can be described as follows:(12)$$\frac{{\mathrm{dC}}_{\mathrm{Pt}\left(\mathrm{IV}\right)}}{\mathrm{dt}}=-k_{1,\mathrm{obs}}\times C_{\alpha -\mathrm{Pt}\left(\mathrm{IV}\right)}-k_{2,\mathrm{obs}}\times C_{\beta -\mathrm{Pt}\left(\mathrm{IV}\right)}$$
under the condition
(13)$$C_{0,\mathrm{Pt}\left(\mathrm{IV}\right)}=C_{t,\alpha -\mathrm{Pt}\left(\mathrm{IV}\right)}+C_{t,\beta -\mathrm{Pt}\left(\mathrm{IV}\right)}$$

In Equations (12) and (13), *k*_1,obs_ and *k*_2,obs_ denote the observed rate constants,$\;C_{0,\mathrm{Pt}\left(\mathrm{IV}\right)}$—the initial concentration of Pt(IV) ions in the reacting solution, and $C_{t,\alpha ,\beta -\mathrm{Pt}\left(\mathrm{IV}\right)}$—concentration of the Pt(IV) ions at time “t”.

In our experiments, mostly isolation conditions (i.e., great excess of the reductant compared to metal ions, changed in the range 50–100) were applied as follows:(14)$$C_{H_{2}\mathrm{Asc}}\gg C_{\mathrm{Pt}\left(\mathrm{IV}\right)}$$

We can assume that during the reduction process, the concentration of ascorbic acid is constant, and the values of the observed rate constant can be described as follows:(15)$$k_{1,\mathrm{obs}}=k_{1}\times C_{0,H_{2}\mathrm{Asc}}$$
where *k*_1_—second-order rate constant, and $C_{0,H_{2}Asc}$—total concentration of ascorbic acid.

The solution of Equation (12) has the following form:(16)$$C_{\mathrm{Pt}\left(\mathrm{IV}\right)}=C_{\alpha -\mathrm{Pt}\left(\mathrm{IV}\right)}\exp(-k_{1,\mathrm{obs}}\times t)+C_{\beta -\mathrm{Pt}\left(\mathrm{IV}\right)}\exp(-k_{2,\mathrm{obs}}\times t)$$

Equation (16) fits well the obtained data shown in Figure 2. However, in the practice, we measure the value of absorbance (not direct concentration); thus we need to modify Equation (16) and take into account the fact that Pt(II) ions in the form of [PtCl_4_]^2−^ also give an absorbance signal at 360 nm [22] (more details given in Section 3.11). Therefore, the final equation has the following form:(17)$$A_{360\;\mathrm{nm}}=A_{1}\times \exp(-k_{1,\mathrm{obs}}\times t)+A_{2}\times \exp(-k_{2,\mathrm{obs}}\times t)+y_{0}$$
where the values A_1,2_ ∝ $C_{\alpha ,\beta -\mathrm{Pt}\left(\mathrm{IV}\right)}$, $y_{0}$—absorbance signal coming from [PtCl_4_]^2−^.

### 3.4. Stoichiometry

To determine the stoichiometry, the reagents were mixed at different molar ratios. The collapse of the graph showing the absorbance as a function of the molar concentration ratio C_0,H2Asc_:C_0,Pt(IV)_ appeared at the ratio of 1:1 (Figure 3). This is in good agreement with the findings of Lemma et al. [7]. It seems that the stoichiometry of the reduction reaction between Pt(IV) and L-ascorbic acid does not change in the presence of oxygen dissolved in the solution. However, it is interesting that the value of the absorbance has not reached zero.

The observation of the obtained color of the solution after mixing the reagents at a ratio of 1:1 (yellow color of the solution) allows us to conclude that the Pt(IV) ions are reduced to Pt(II). It is worth noting that at a higher value of the molar ratio of ascorbic acid to metal ions, i.e., 2:1, the solution turns colorless, and grey sediment appears with time. This suggests that with time, Pt(II) is reduced to Pt(0).

### 3.5. The Dependency of k_obs_ vs. Initial Concentration of the Reductant

In order to determine the correlation between the reaction rate constant and the excess of the reductant, kinetic measurements were conducted for the following excess concentrations of ascorbic acid compared to the Pt(IV) ions: 40, 50, 60, 70, 80, and 100 times. The determined average of the observed and second-order values of the rate constants are summarized in Table 2.

By plotting the dependence of the rate constant as a function of the initial concentration of the reductant, the values of the second-order rate constants *k*_1_ and *k*_2_ can be determined. The linear relationship (Figure 4a), as well as the fact that it has its origin at (0, 0), confirm the correctness of the adopted expression for the observed rate constant, *k*_1,obs_ (15).

The value of the second-order rate *k*_1_ has the value of the directional coefficient from the fit to the relationship *k*_1,obs_ =f(C_0,H2Asc_), and it equals 5.15 M^−1^s^−1^. In the case of the second observed rate constant (*k*_2,obs_), there is no such linear dependency (Figure 4b). The value of the observed rate is constant. This suggests that the process is independent of the initial concentration of ascorbic acid. 

### 3.6. The Influence of Temperature on Second-Order Rate Constant

In order to determine the activation parameters like energy (E_A_), enthalpy (ΔH^≠^) and entropy ΔS^≠^ for the reaction between the Pt(IV) ions and the reductant, the kinetic study was carried out in different temperature ranges (from 20 °C to 40 °C). 

Using the logarithmic form of the Arrhenius equation and the logarithmic form of the Eyring equation, the values of energy, enthalpy and entropy of activation were determined graphically. The experimental results of the observed reaction rate constants, as a function of temperature, were plotted in an appropriate system, i.e., ln(*k*_1,2_) vs. 1/T (Figure 5a, Appendix A) and ln(*k*_1,2_/T) vs. 1/T (Figure 5b, Appendix A).

The obtained dependencies are linear. Knowing the equations of the straight lines, the relevant parameters were determined in both equations. The values of these parameters, i.e., E_A_ and A in the Arrhenius and values ΔH^≠^ and ΔS^≠^ in the Eyring equation, are given in Table 3 and Table 4.

The analysis of the influence of temperature on the reaction rate constant was carried out for the reaction running at pH = 2.5. Under these conditions, an increase in the rate constant (*k*_1_) with an increase in the temperature was observed. However, an analogical relation was not observed for *k*_2_ (see Appendix A). The lack of influence of the temperature on the second process may indicate that at least two competitive reactions are taking place there. One of them is exothermic, and the other is endothermic. The obtained results suggest that the Arrhenius and Eyring equations are fulfilled for the studied system, but only for the first process. Moreover, the determined value of ΔH^≠^ is the same as that reported by Lemma et al. [7] in the deaerated reacting system (pH = 5.74, I = 1 M), whereas the value of ΔS^≠^ obtained in our study is lower (−73 J/(K × mol)) than that determined by Lemma et al. [7]. The negative activation entropy can be attributed to most of the electron transfers in the transition state. For example, two solvated reagent molecules may represent a single transition state that results in a reduction in the degree of freedom of translation.

### 3.7. The Influence of pH

In order to investigate the influence of the reductant forms on the rate and on the mechanism of the reaction of the Pt(IV) ions, the process of reduction was carried out at different pH levels (2.15–5.04). The obtained observed rate constants are gathered in Table 5.

By analyzing the obtained data, it can be seen that as the pH of the environment increases, the reaction rate of the platinum (IV) complex reduction also increases. This can be explained by the increasing share of the dissociated form of ascorbic acid, which is more reactive [19,35]. As shown in Appendix A, the character of the kinetic curves also changed with the pH. What is especially interesting is the shape of the kinetic curves obtained at pH = 3.65 (see, Appendix A), and the fitted curves shown in Appendix A. Comparing these data (fitting equations), we obtain a good fit for both the single and double exponential equation. This means the rate of the first and the second process are equal (*k*_1,obs_ = *k*_2,obs_, Table 5). At a higher pH, only one process takes place, and it relates to the ${\left[{\mathrm{PtCl}}_{5}\left(H_{2}O\right)\right]}^{-}$ reduction. The increased share of the hydrolyzed form of Pt(IV) can be explained by the hydrolysis progress (2), which strongly depends on the pH [26,36]. 

### 3.8. The Influence of Ionic Strength

In order to determine whether there is a salt effect during the reaction between the Pt(IV) complex and ascorbic acid in the examined system, a different ionic strength was used through the addition of NaClO_4_ in the range from 0 to 0.4 M. The observed rate constants (*k*_1,2,obs_) are summarized in Table 6.

As shown in Table 6, an increase in the applied ionic strength has little effect on the value of the rate constants (see also Appendix A).

### 3.9. The Influence of Chloride Concentration

To examine the influence of the addition of chloride ions on the rate of the Pt(IV) ion reduction, different initial concentrations of sodium chloride were used. The values of the observed and second-order rate constants are gathered in Table 7.

It is shown in Table 7 that with an increase in the content of chloride ions in the range 0.00–0.40 M, the rate of the Pt(IV) ion reduction increases. The values of absorbance obtained from the fit of equations to the kinetic curves (see, Appendix A) are also interesting. Based on the values y_0_, A_1_ and A_2_ (16), and the known values of the molar coefficient for the platinum species, it was possible to determine the exact concentration of each species. The share between them is shown in Figure 6.

The obtained results indicate that with an increase in the chloride ions, an increase in the amount of [PtCl_6_]^2−^ ions can be expected, because the addition of chloride ions reverses hydrolysis process (2). Thus, the amount of [PtCl_5_(H_2_O)]^−^ ions in the solution decreases. The obtained values of [PtCl_4_]^2−^ ([Cl^−^] = 0.0 M, see Appendix A) are interesting since they exceed the value of the initial concentration of Pt(IV), i.e., 0.1 mM. This can be explained by the presence of Pt(0) and is described in detail in Section 3.11. Also, chloride ions are reactants in reactions (6) and (8), and might lead to the formation of ${\left[{\mathrm{PtCl}}_{5}\left(H_{2}O\right)\right]}^{-}$ (7) and ${\left[{\mathrm{PtCl}}_{6}\right]}^{2-}$ ions (8).

### 3.10. The Role of Oxygen Dissolved in Aqueous Solution in the Reduction Process

In order to understand the role of oxygen in the process of Pt(IV) reduction using ascorbic acid, we compared the kinetic curves obtained in a standard process (containing oxygen) and in deaerated conditions (each solution was purged with nitrogen for 30 min. before mixing).

As expected, the kinetic curves differ depending on the conditions (see Figure 7a). The change in the character of the kinetic curves suggests that one of the processes was eliminated (see Figure 7b) or the overall process accelerates in the deaerated solution (see comparison of fitting curve for both experimental conditions with oxygen and in deaerated solution fitted using the same fitting equation, shown in Appendix A). 

The obtained results (Figure 7a) show that using even smaller amounts of reductant in the case of the deaerated solution leads to a process acceleration. The value of the observed rate constant (*k*_1,obs_) was faster by two times, and the value of *k*_2,obs_ is ~six times higher compared to the rate constants obtained for the solution containing oxygen, whereas at the same reductant content, the value of *k*_1,obs_ is the same, and the value of *k*_2,obs_ is ~three times higher (see Appendix A) compared to the solution containing oxygen. This suggests that when oxygen is present in the solution, a deprotonated form of ascorbic acid is consumed (4, 5). The obtained results indicate that a smaller ascorbic acid amount is more effective. This observation can be explained by the dissociation process of the reductant, for which the dissociation degree increases with the dilution (see Appendix A). A higher value of dissociation degree means that in the reacting system, in practice, we have five times more HAsc^−^ than in the solution with a higher reductant concentration (0.06 M) (see Appendix A). 

Another interesting observation is that the kinetic curve for the studied solution reaches a certain value. In the case of the deaerated solution, the recorded level of absorbance is 0.22 (see, Figure 7b), and it suggests the presence of Pt(II) ions, which was also observed by Senapati et al. [37]. They indicated that the square planar Pt(II) complex has a characteristic spectrum in the UV–Vis wavelength range. Moreover, the appearance of a solid phase in the system, i.e., platinum nanoparticles, can also influence the absorbance level. The presence of the solid phase causes turbidity of the tested solution and raises the spectrum in the entire wavelength range. 

### 3.11. Mechanism Description 

Based on the obtained results, we suggest that the reduction process of Pt(IV) ions to Pt(0) generally proceeds along two reaction routes, which are sensitive to pH, temperature and oxygen presence. These factors have many implications that are responsible for the reaction chain and complexity of the reduction–oxidation mechanism of the components. The presence of oxygen in the aqueous solution is especially responsible for ascorbic acid oxidation and reacting species formation like hydrogen peroxide, hypochlorous acid and chlorine [22]. It also seems that the irradiation of the reacting solution at 360 nm during the reduction also has consequences in the starting Pt(IV) species. According to Cox et al. [30], during irradiation at 365 nm, the photoaquation of hexachloroplatinate(IV) takes place and has a redox nature. 

Taking into account the used pH value (mainly 2.5) and the registered spectrum intensity (Figure 1a), at the beginning of the process, we have no hydrolyzed form of Pt(IV) (2). 

However, the irradiation at 360 nm can also affect the rate of the hydrolysis process [38]. Thus, under the experimental conditions, we expected the coexistence of both Pt(IV) species, whose share depends on the pH, chloride content and temperature. 

The reductant of one form of ascorbic acid is dominating (pH = 2.5). However, both H_2_Asc and HAsc^−^ (3) are responsible for the Pt(IV) reduction. The share between these forms depends on the pH, temperature and dissociation degree of the solution and has a strong influence on the rates. Hence, the reduction process of Pt(IV) to Pt(II) can be described as follows:(18)$${\left[{\mathrm{PtCl}}_{6}\right]}^{2-}+H_{2}\mathrm{Asc}\overset{}{\to }{\left[{\mathrm{PtCl}}_{4}\right]}^{2-}+\mathrm{DHA}+2H^{+}+2{\mathrm{Cl}}^{-}$$
(19)$${\left[{\mathrm{PtCl}}_{6}\right]}^{2-}+2{\mathrm{HAsc}}^{-}\overset{}{\to }{\left[{\mathrm{PtCl}}_{4}\right]}^{2-}+2\mathrm{DHA}+2{\mathrm{Cl}}^{-}$$
(20)$${\left[{\mathrm{PtCl}}_{5}\left(H_{2}O\right)\right]}^{-}+H_{2}\mathrm{Asc}\overset{}{\to }{\left[{\mathrm{PtCl}}_{3}\left(H_{2}O\right)\right]}^{-}+\mathrm{DHA}+2H^{+}+2{\mathrm{Cl}}^{-}$$
(21)$${\left[{\mathrm{PtCl}}_{5}\left(H_{2}O\right)\right]}^{-}+2{\mathrm{HAsc}}^{-}\overset{}{\to }{\left[{\mathrm{PtCl}}_{3}\left(H_{2}O\right)\right]}^{-}+2\mathrm{DHA}+2{\mathrm{Cl}}^{-}$$

The observed rate constant (*k*_1*,obs*_) depends on the initial reductant concentration, whereas *k*_2,obs_ is independent (Figure 4b). This suggests that another process(es) should be taken into account, e.g., (22)–(26).
(22)$${\left[{\mathrm{PtCl}}_{4}\right]}^{2-}+\mathrm{HOCl}+H^{+}\to {\left[{\mathrm{PtCl}}_{5}\left(H_{2}O\right)\right]}^{-}$$
(23)$${\left[{\mathrm{PtCl}}_{5}\left(H_{2}O\right)\right]}^{-}+{\mathrm{Cl}}_{2}+{\mathrm{Cl}}^{-}+{\left[{\mathrm{PtCl}}_{4}\right]}^{2-}\to 2{\left[{\mathrm{PtCl}}_{6}\right]}^{2-}+H_{2}O$$
(24)$${\left[{\mathrm{PtCl}}_{4}\right]}^{2-}+H_{2}O_{2}\to \mathrm{trans}-{\left[{\mathrm{PtCl}}_{4}{\left(\mathrm{OH}\right)}_{2}\right]}^{2-}$$
(25)$${\left[{\mathrm{PtCl}}_{4}\right]}^{2-}+{\mathrm{Cl}}_{2}\to {\left[{\mathrm{PtCl}}_{6}\right]}^{2-}$$
(26)$$2{\left[{\mathrm{PtCl}}_{4}\right]}^{2-}↔{\left[{\mathrm{PtCl}}_{6}\right]}^{2-}+2{\mathrm{Cl}}^{-}+\mathrm{Pt}\left(0\right)$$

Reactions (22)–(25) are possible if reaction (3) generates a dissociated form of ascorbic acid, which reacts with the oxygen dissolved in the aqueous solution (4) and (5). At our conditions, reaction (5) is the most probable, which is perhaps catalyzed by the presence of the transition metal, i.e., Pt(II) ions via the analogy to Cu(II) [33], produced in reactions (18) and (19).
(27)$${\mathrm{HAsc}}^{-}+O_{2}+H^{+}\overset{\mathrm{Pt}\left(\mathrm{II}\right)\;\mathrm{as}\;\mathrm{catalyst}}{\to }\mathrm{DHA}+H_{2}O_{2}$$

Moreover, in our reacting system, reaction (26) can be present both in the reacting system containing oxygen and in the deaerated solution. The confirmation of disproportionation (26) can be the fact that the absorbance value on the kinetic curve does not reach zero (see Appendix A), even at a longer time. The registered absorbance level can be related to the presence of Pt(II) ions in the solution and the appearance of the metallic phase. To prove this, we performed a certain calculation. Taking into account [22], the value of the molar coefficient at 390 nm for [PtCl_4_]^2−^ equals ε_390 nm_ = 56 M^−1^cm^−1^; thus, in our experimental condition, if 100% of the Pt(IV) goes to Pt(II), the value of the absorbance should be A_360 nm,Pt(II)_ = 0.056. Taking into account that the characteristic peak on the [PtCl_4_]^2−^ spectrum is broad (see [22]), we assumed that the value of the molar coefficient at 360 nm is similar to that registered at 390 (ε_360 nm_
≈ ε_390 nm_). Based on this assumption and the kinetic curve (see Appendix A), we can calculate the value of the absorbance according to the following relation:(28)$$\Delta A_{t=6\;\min.}=(A_{360\;\mathrm{nm}}-A_{550\;\mathrm{nm}})-(A_{360\;\mathrm{nm},\;Pt\left(II\right)})=0.108$$
where $A_{360\;\mathrm{nm}}$—the value of the absorbance registered at 360 nm and $A_{550\;\mathrm{nm}}$—the absorbance correction related to the turbidity level registered at 550 nm (0.025, see Appendix A). 

The appearance of a solid phase in the system, i.e., platinum nanoparticles, can also influence the absorbance level. The presence of the solid phase causes the turbidity of the tested solution and raises the spectrum in the entire wavelength range. Thus, such an absorbance correction is necessary in order to extract the “absorbance signal” from the platinum ions.

From Equation (28) and by knowing the value of ε_390_ nm for the Pt(IV) ions, we can calculate the concentration of this compound (2.2 × 10^−4^ M) at time t = 6 min., and the concentration of the Pt(II) ions, which is equal to 7.8 × 10^−4^ M. Again, the calculation of the absorbance value based on these data gives the value of 0.044, which is lower than the theoretical value, i.e., 0.056. This difference may suggest that in the reacting solution, we have one more Pt(II) species, which coexists with Pt(0), [PtCl_4_]^2−^ and [PtCl_6_]^2−^. For example, it can be a hydrolyzed form of Pt(II). The presence of Pt(0) and PtNPs might also have catalytic consequences in the process of H_2_O_2_ decomposition [39,40]. However, the PtNPs are produced at a longer time, as shown in Appendix A. The reduction process of Pt(IV) to Pt(II) ions in the deaerated solution is less complex, as shown in Section 3.10. The obtained character of the kinetic curve (Figure 7b) shows that the process can be limited to reactions (18)–(21), at which the reduction of [PtCl_6_]^2−^ is slower than the [PtCl_5_(H_2_O)]^−^ species. As a next step (which is not described here in detail), reactions (29) and (30) take place, and platinum nanoparticles are formed.
(29)$${\left[{\mathrm{PtCl}}_{4}\right]}^{2-}+H_{2}\mathrm{Asc}\to \mathrm{Pt}\left(0\right)+\mathrm{DHA}+2H^{+}+4{\mathrm{Cl}}^{-}$$
(30)$${\left[{\mathrm{PtCl}}_{4}\right]}^{2-}+2{\mathrm{HAsc}}^{-}\to \mathrm{Pt}\left(0\right)+2\mathrm{DHA}+4{\mathrm{Cl}}^{-}$$

## 4. Conclusions

The reduction of Pt(IV) ions using ascorbic acid leads to the formation of Pt(II) ions, which are next reduced to Pt(0), and finally, platinum nanoparticles are formed. The process carried out in the solution containing oxygen leads to many complications related to the reactive species formation, which coexists with ascorbic acid as the main reducing agent. From the experimentally obtained kinetic curves, the rate constants of the reaction at different conditions (concentrations of reactants and chloride ions, pH, temperature and ionic strength) were determined. For the studied system, the values of the enthalpy and entropy of activation were determined to be 52 kJ/mol and −73 kJ/(mol × K), respectively. 

The obtained kinetic curves allow us to assume that the observed pseudo-first rate constant is responsible for the rate of the reduction reaction of Pt(IV) to Pt(II) in (18)–(21), whereas the second observed pseudo-first rate constant is responsible for the slower process. Among them, reactions (22)–(26) are considered. The formed [PtCl_4_]^2−^ is a particularly reactive compound, because it oxidizes in the presence of hydrogen peroxide, hypochlorous acid and chlorine. Each of these reagents can be formed in our system as a consequence of the reaction between the deprotonated form of ascorbic acid and oxygen dissolved in the solvent. Moreover, the presence of the oxygen in the solution strongly influences the amounts of dissociated and more reactive forms of ascorbic acid. The obtained results show that the mechanism of the Pt(IV) to Pt(II) reduction changes either in the presence of oxygen or with a pH above 3.6. In the first case, reactions (4), (5), (22)–(25) and (27) are eliminated. In the second case, the amount of [PtCl_4_]^2−^ is limited by hydrolysis process (2) and increases the amount of HAsc^−^ as a more active reducing compound. These aspects should be taken into account in the process of platinum nanoparticle synthesis and for a more comprehensive description of the mechanism.

## Figures and Tables

**Figure 1 materials-16-04630-f001:**
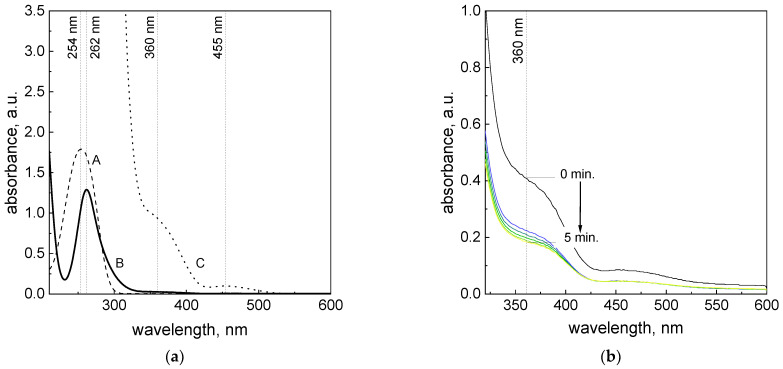
UV–Vis spectrum of ascorbic acid (A) and Pt(IV) ions in H_2_O (B, C) (**a**). Conditions: C_0,H2Asc_ = 0.2 mM, C_0,Pt(IV)_ = 0.05 mM (B); C_0,Pt(IV)_ = 2.0 mM (C). Optical path length—1 cm, T = 20 °C. Spectrum evolution observed during reduction of Pt(IV) ions using ascorbic acid (**b**). Conditions: C_0,Pt(IV)_ = 1.0 mM, C_0,H2Asc_ = 60 mM, pH = 2.50 ± 0.05, T = 313 ± 0.1 K, I = 0.06 M.

**Figure 2 materials-16-04630-f002:**
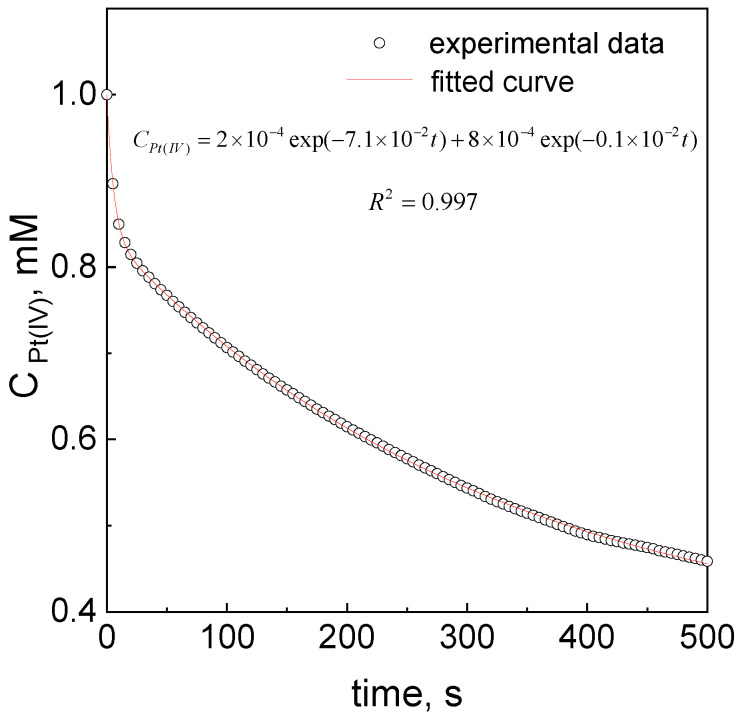
The sample of kinetic curve for Pt(IV) ion reduction using ascorbic acid. Conditions: C_0,Pt(IV)_ = 1.0 mM, C_0,H2Asc_ = 60 mM, pH = 2.5 ± 0.5, T = 313 ± 0.1 K, I = 0.06 M.

**Figure 3 materials-16-04630-f003:**
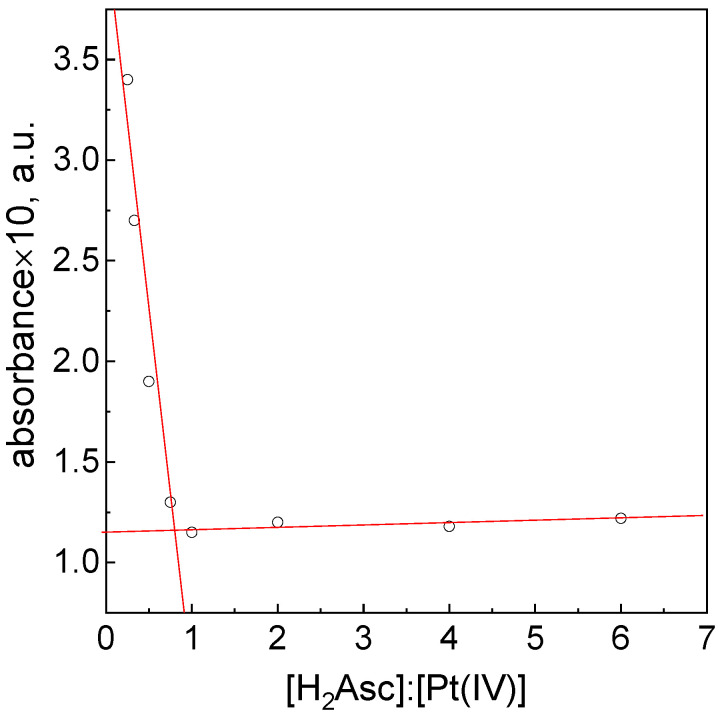
The dependency of absorbance vs. C_0,H2Asc_:C_0,Pt(IV)_ ratio. Conditions: pH = 2.50 ± 0.05, T = 313 ± 0.1 K, I = 0.06 M.

**Figure 4 materials-16-04630-f004:**
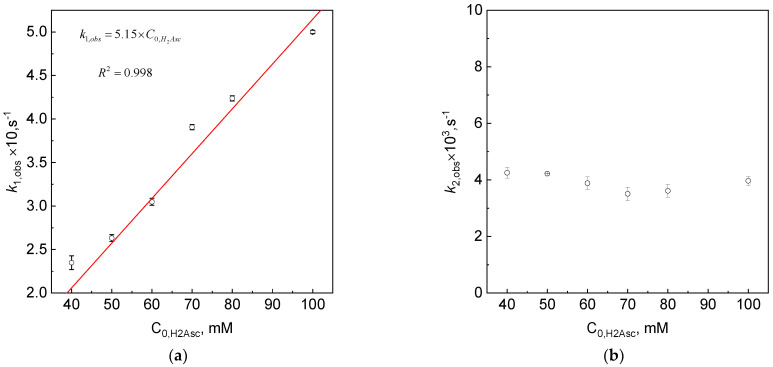
Dependency of observed rate constant as a function of initial reductant concentration for first (**a**) and second process (**b**). Conditions: C_0,Pt(IV)_ = 1.0 mM, C_0,H2Asc_ = 40–100 mM, pH = 2.50 ± 0.05, T = 313 ± 0.1 K, I = 0.4 M.

**Figure 5 materials-16-04630-f005:**
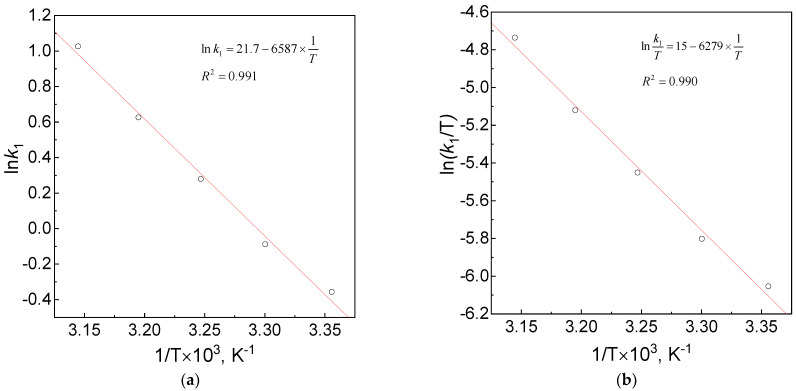
Arrhenius (**a**) and Eyring (**b**) dependency for reduction of Pt(IV) ions using ascorbic acid. Conditions: C_0,Pt(IV)_ = 1.0 mM, C_0,H2Asc_ = 60 mM, pH = 2.50 ± 0.05, I = 0.06 M.

**Figure 6 materials-16-04630-f006:**
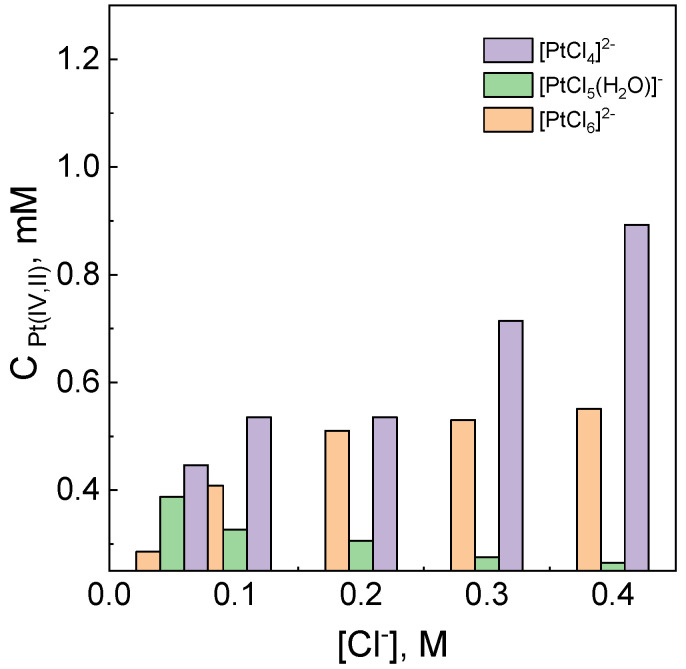
The share of the platinum species present in the solution on the chloride ion concentration. Conditions: C_0,Pt(IV)_ = 1.0 mM, C_0,H2Asc_ = 60 mM, pH = 2.50 ± 0.05, T = 313 K.

**Figure 7 materials-16-04630-f007:**
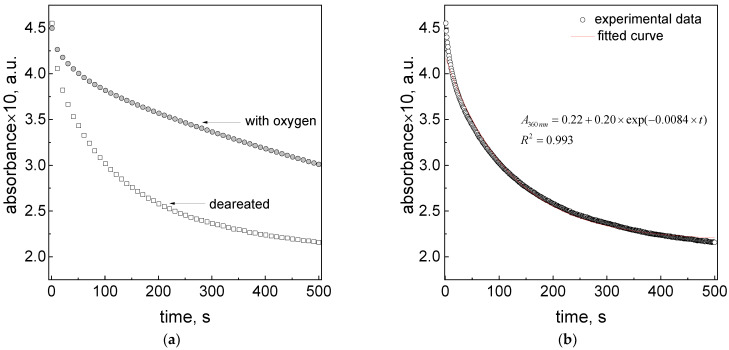
Kinetic curves registered at 360 nm for solutions containing Pt(IV) using ascorbic acid, saturated oxygen and deaerated solution (**a**); experimental kinetic data and fitted curve obtained for deaerated solution (**b**). Conditions: C_0,Pt (IV)_ = 1.0 mM, C_0,H2Asc_ = 0.06 M; C_0,H2Asc_ = 2.0 mM (deaerated solution), T = 313 ± 0.1 K, pH = 2.50 ± 0.5.

**Table 1 materials-16-04630-t001:** Conditions of conducted experiments aimed at studying the kinetics of the reaction of reducing Pt(IV) chloride complexes using L-ascorbic acid.

Initial Concentration of Reagents		Ionic Strength	T	pH
C_0,Pt(IV)_, mM	C_0,H2Asc_, M	I, M	K	
The stoichiometry				
C_0,H2Asc_:C_0,Pt(IV)_ 0.25 0.33 0.5 0.75 1 2 4 6		0.10	313	2.50



Dependency of *k*_obs_ vs. initial reductant concentration				
1.00	0.04	0.40	313	2.50
	0.05			
	0.06			
	0.07			
	0.08			
	0.10			
The influence of temperature				
1.00	0.06	0.06	287	2.50
			298	
			303	
			308	
			313	
The influence of pH				
1.00	0.06	0.06	313	2.15 2.50
				3.65
				4.17
				5.04
The influence of ionic strength (addition of NaClO_4_) on the rate constants				
1.00	0.06	0.00 0.05 0.10 0.20 0.30 0.40	313	2.50
The influence of chloride ions on the rate constants (at constant value of [Na^+^] = 0.4 M)				
1.00	0.06	0.4	313	NaCl addition 0.00 0.05 0.10 0.20 0.30 0.40

**Table 2 materials-16-04630-t002:** Effect of reductant excess on pseudo-first and second-order reaction rate constants along with the deviation (of 6 replicates). Conditions: C_0,Pt (IV)_ = 1.0 mM, pH = 2.50 ± 0.05, T = 313 ± 0.1 K, I = 0.4 M.

$C_{0,H_{2}\mathrm{Asc}}$, mM	$k_{1,\mathrm{obs}}\;\times \;10^{2}$ s^−1^	*k*_1_ M^−1^s^−1^	$k_{2,\mathrm{obs}}\;\times \;10^{3}$ s^−1^	$k_{2}\;\times \;10^{2}$ M^−1^s^−1^
40	23.49 ± 0.79	3.91 ± 0.13	4.25 ± 0.19	7.08 ± 0.32
50	26.32 ± 0.40	4.38 ± 0.07	4.22 ± 0.04	7.04 ± 0.37
60	30.49 ± 0.40	5.08 ± 0.07	3.89 ± 0.22	6.48 ± 0.40
70	39.06 ± 0.32	5.51 ± 0.05	3.51 ± 0.24	5.85 ± 0.40
80	42.37 ± 0.30	7.06 ± 0.05	3.61 ± 0.23	6.02 ± 0.38
100	50.00 ± 0.20	8.33 ± 0.03	3.97 ± 0.16	6.61 ± 0.27

**Table 3 materials-16-04630-t003:** List of constant parameters in the Arrhenius equation for the reaction between the Pt(IV) ions and ascorbic acid. Conditions: C_0,Pt(IV)_ = 1.00 mM, C_0,H2Asc_ = 60.0 mM, pH = 2.50 ± 0.05, I = 0.06 M.

ln A	$A\;\times \;10^{-7}$ [M^−1^s^−1^]	E_A_/R	E_A_ $\left[\mathrm{KJ}\;\times \;{\mathrm{mol}}^{-1}\right]$
21.7	26	6587	54

**Table 4 materials-16-04630-t004:** List of constant parameters in the Arrhenius equation for the reaction between the Pt(IV) complex and ascorbic acid. Conditions: C_0,Pt(IV)_ = 1.00 mM, C_0,H2Asc_ = 60.0 mM, pH = 2.50 ± 0.05, I = 0.06 M.

ΔH/(R)	ΔH^±^ [kJ × mol^−1^]	(23.77 + Δs/R)	∆S^±^ [J × K^−1^ × mol^−1^]
6279	52	15	−73

**Table 5 materials-16-04630-t005:** Influence of the pH of the environment on the values of the observed rate constants (*k*_1,2, obs_) with the standard deviation (mean of 5 replicates). Conditions: C_0,Pt(IV)_ = 1.00 mM, C_0,H2Asc_ = 60.0 mM, T = 313 ± 0.1 K, I = 0.06 M.

pH	$k_{1,\mathrm{obs}}\;\times \;10^{-2}$ s^−1^	$k_{2,\mathrm{obs}}\;\times \;10^{3}$ s^−1^
2.15 ± 0.05	10.86 ± 0.06	1.20 ± 0.06
2.50 ± 0.05	16.76 ± 0.22	0.66 ± 0.18
3.65 ± 0.05	62.89 ± 1.02	62.89± 0.18
4.17 ± 0.05	89.28 ± 0.03	-
5.04 ± 0.05	105.2 ± 0.10	-

**Table 6 materials-16-04630-t006:** The values of the observed reaction rate constants along with the standard deviation (mean value of 5 replicates) obtained at different ionic strengths. Conditions: C_0,Pt(IV)_ = 1.0 mM, C_0,H2Asc_ = 60.0 mM, T = 313 ± 0.1 K, pH = 2.50 ± 0.05.

I M	$k_{1,\mathrm{obs}}\;\times \;10^{2}$ s^−1^	$k_{2,\mathrm{obs}}\;\times \;10^{3}$ s^−1^
0.00	17.45 ± 0.06	3.30 ± 0.02
0.05	25.00 ± 0.02	2.11 ± 0.05
0.10	27.78 ± 0.02	2.36 ± 0.03
0.20	29.76 ± 0.02	3.44 ± 0.05
0.30	29.82 ± 1.31	3.52 ± 0.19
0.40	30.49 ± 0.40	3.89 ± 0.22

**Table 7 materials-16-04630-t007:** The influence of chloride ion concentration on the values of the observed (*k*_1,2, obs_) and second-order rate constants (*k*_1,2_) with the standard deviation (mean of 5 replicates). Conditions: C_0,Pt(IV)_ = 1.00 mM, C_0,H2Asc_ = 60.0 mM, [Na^+^] = 0.4 M, T = 313 ± 0.1 K, pH = 2.50 ± 0.05, I = 0.4 M.

[Cl^−^] M	$k_{1,\mathrm{obs}}\;\times \;10^{2}$ s^−1^	*k*_1_ M^−1^s^−1^	$k_{2,\mathrm{obs}}\;\times \;10^{3}$ s^−1^	$k_{2}\;\times \;10^{2}$ M^−1^s^−1^
0.00	30.48 ± 0.40	5.08	3.89 ± 0.05	6.48
0.05	39.06 ± 0.30	6.51	4.41 ± 0.03	7.34
0.10	35.84 ± 0.39	5.97	4.52 ± 0.05	7.54
0.20	37.45 ± 0.10	6.24	5.21 ± 0.06	8.68
0.30	40.32 ± 0.20	6.72	5.57 ± 0.06	9.41
0.40	40.65 ± 0.20	6.77	5.41 ± 0.10	9.01

## Data Availability

Not applicable.

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
