# Peer review of "The Kinetics of the Redox Reaction of Platinum(IV) Ions with Ascorbic Acid in the Presence of Oxygen"

_materials, 2023, doi:10.3390/ma16134630_

Round 1

Reviewer 1 Report

This paper reports the kinetics of redox reaction of platinum(IV) ions with ascorbic acid at different reagents concentration and environment condition. During reduction process, the presence of the oxygen leads to the process of Pt(IV) reduction become more complex, especially in mild acidic medium. The rate constants and kinetic equations drawn from the kinetic trace obtained by spectrophotometry. Based on the values of enthalpy and entropy of activation, the mechanism of Pt(IV) reduction was proposed. I find the study well carried out, interesting, and suitable for Materials. There are some points, however, I would like to see clarified before publication.

1)      In Figure 2, the kinetic curve for Pt(IV) ions reduction with ascorbic acid at the molar concentration ratio of 1:60, while Pt(â…¡) could be reduced to Pt(0). The oxygen not only affects Pt(â…¡) , but also affects Pt(0).  I suggest the authors add the kinetics of redox reaction of platinum(IV) ions with ascorbic acid at the molar concentration ratio of 1:1 in the presence of inert gas and oxygen.

2)      In Figure 6, the amount of [PtCl4]2- is more than [PtCl6]2-and [PtCl5(H2O)]-. Why the characteristic band of [PtCl4]2- couldn’t find in UV-Vis spectrum and how to determine its existence?

3)      Is the R2 in Figure 4a better than Figure 5?

4)      Here are some minor comments.

‘[PtCl6]’ should be rewritten as ‘[PtCl6]2-

Line 93. ‘3.1. Spectra of reagents’, should be rewritten as ‘3.2. Spectra of reagents

The format of references needs to be unified

Author Response

Review 1

Comments and Suggestions for Authors

This paper reports the kinetics of redox reaction of platinum(IV) ions with ascorbic acid at different reagents concentration and environment condition. During reduction process, the presence of the oxygen leads to the process of Pt(IV) reduction become more complex, especially in mild acidic medium. The rate constants and kinetic equations drawn from the kinetic trace obtained by spectrophotometry. Based on the values of enthalpy and entropy of activation, the mechanism of Pt(IV) reduction was proposed. I find the study well carried out, interesting, and suitable for Materials. There are some points, however, I would like to see clarified before publication.

  • In Figure 2, the kinetic curve for Pt(IV) ions reduction with ascorbic acid at the molar concentration ratio of 1:60, while Pt(â…¡) could be reduced to Pt(0). The oxygen not only affects Pt(â…¡) , but also affects Pt(0). I suggest the authors add the kinetics of redox reaction of platinum(IV) ions with ascorbic acid at the molar concentration ratio of 1:1 in the presence of inert gas and oxygen.

Reply: Thank you for valuable comments. The suggestion of the Reviewer is very important. We wanted to add this experiment. Unfortunately, waiting for the stopped flow repair, delayed delivery of replacement components, we were unable to implement it. However, conditions close to those recommended by the Reviewer are shown in the paragraph on the role of oxygen. We showed there (paragraph 3.10, Fig. 7) kinetic curve obtained at molar ratio 1:2, and also kinetic curves fitting has been showed in SM (Fig. S5 b, c). The fitted curves are different, thus the process is also different. We also agree with the Reviewer, that the presence of oxygen influenced on the Pt(II) reduction to Pt(0). For this reason, we plan to continue our studies, related to the next step, i.e. reduction Pt(II) ions to metallic phase and these studies will be also focused on more deeply structure characterisation of obtained platinum nanoparticles.

  • In Figure 6, the amount of [PtCl4]2- is more than [PtCl6]2-and [PtCl5(H2O)]-. Why the characteristic band of [PtCl4]2-couldn’t find in UV-Vis spectrum and how to determine its existence?

Reply: The characteristic band of Pt(II) is less intense comparing to platinum others forms, especially Pt(IV). This fact can be confirmed by the value of molar coefficient determined for this compound and according to Murray et al (Dalton Trans. 2014, 43, 6308 – 6314) equals 56 M-1cm-1 at 390 nm. Moreover, the location of characteristic peak is close to that characteristic for Pt(IV) species like[PtCl6]2-and [PtCl5(H2O)]- which is located at 353 nm (Dalton Trans. 2014, 43, 6308 – 6314). Moreover, these species have much higher value of molar coefficient, i.e. 490 M-1cm-1. The existence of Pt(II) compound can be calculated from the UV-Vis spectra, but only if no Pt(IV) species and metallic phase (PtNPs) are present in the solution. In the case, when next to Pt(II) another Pt(IV) compounds are presents, the UV-Vis spectra overlap and it is impossible to distinguish exact amount of each species.

  • Is the R2 in Figure 4a better than Figure 5?

Reply: Yes.

4)      Here are some minor comments.

‘[PtCl6]’ should be rewritten as ‘[PtCl6]2-

Line 93. ‘3.1. Spectra of reagents’, should be rewritten as ‘3.2. Spectra of reagents

The format of references needs to be unified

Reply: According to Reviewer’s comments, these errors were corrected.

Reviewer 2 Report

Report on the manuscript materials-2347065; entitled “The kinetics of the redox reaction of Pt(IV) ions with ascorbic acid in the presence of oxygen”.

The submitted review should be revised. The following points should be addressed:

1. The submitted manuscript should be revised to be free from editing or grammar errors.

2. In experimental work, the estimation of k(obs) should be mentioned in addition to the expected products of equation 1 could be mentioned.

3. The authors wrote that “where k1 – first order rate constant” and the unit and meaning of this constant is like the second order not first, please, revise it! (In table 2, the authors have the same mistake).

4. “It seems that stoichiometry of reaction reduction between

Pt(IV) and L-ascorbic acid does not change in the presence of oxygen dissolved in the solution. However, interesting is that the value of absorbance has not reached zero value.”, please, continue the discussion to understand what is the interesting and why not reached to zero?

5. “the solution turns colourless and with time grey sediment appears. It suggests, that Pt(II) is reduced to Pt(0) with time”, the authors should confirm this conclusion by measuring the conductivity and more suitable references way from self-citations.

6. More discussion about thermodynamic parameters should be supported.

7. “an increase in the applied ionic strength has little effect on the value of the rate constants” still there is slight increase so, why the increase of ionic strength enhances the reaction rate, is this related to the formation of more hydrophilic intermediate?

8. In mechanism, the authors suggested the photolysis step in number 2, the authors don’t write any thing in the experimental work about the control of light, is light effect was studied or no?

9. the comparison between sodium borohydride and ascorbic acid should be clear in terms of rate constants and thermodynamic parameters in addition to suggested mechanism.

The submitted manuscript should be revised to be free from editing or grammar errors. 

Author Response

Review 2

Comments and Suggestions for Authors

Report on the manuscript materials-2347065; entitled “The kinetics of the redox reaction of Pt(IV) ions with ascorbic acid in the presence of oxygen”.

The submitted review should be revised. The following points should be addressed:

  1. The submitted manuscript should be revised to be free from editing or grammar errors.

Reply: According to Reviewer’s suggestion, the manuscript was corrected. The all corrections were made in tracking mode and attached to Cover Letter.

  1. In experimental work, the estimation of k(obs) should be mentioned in addition to the expected products of equation 1 could be mentioned.

Reply: According to Reviewer’s suggestion, the reaction (1) was supplemented by the rate constant k, which appeared in the equation. We decided to introduce value of k instead of kobs at this place, because, kobs relates to “isolation conditions” and this term was introduced to the text on the page 7.

  1. The authors wrote that “where k1 – first order rate constant” and the unit and meaning of this constant is like the second order not first, please, revise it! (In table 2, the authors have the same mistake).

Reply: Thank you for this comment. It should be second-order rate and we checked all manuscript and necessary changes were made in the text and tables description.

  1. “It seems that stoichiometry of reaction reduction between Pt(IV) and L-ascorbic acid does not change in the presence of oxygen dissolved in the solution. However, interesting is that the value of absorbance has not reached zero value.”, please, continue the discussion to understand what is the interesting and why not reached to zero?

Reply: That is right, that stoichiometry is independent on the oxygen presence. The value of absorbance do not reach zero and it is related to the “signal” coming from Pt(II), and then from metallic form. More details were gathered in paragraph in 3.11.

  1. “the solution turns colourless and with time grey sediment appears. It suggests, that Pt(II) is reduced to Pt(0) with time”, the authors should confirm this conclusion by measuring the conductivity and more suitable references way from self-citations.

Reply: According to Reviewer’s suggestion, we made additionally tests, i.e. Tyndall effect, which has been positive and we used DLS method for size determination (polydisperse sample, size: a few um). We believe, that applied techniques are acceptable for the Reviewer. These studies will be continued and more detailed structure analysis will be performed.

  1. More discussion about thermodynamic parameters should be supported.

Reply: Discussion about thermodynamics parameters was supported by cited references.

  1. “an increase in the applied ionic strength has little effect on the value of the rate constants” still there is slight increase so, why the increase of ionic strength enhances the reaction rate, is this related to the formation of more hydrophilic intermediate?

Reply: At this stage it is difficult to confirm. However, slight increase can be related with the more hydrophilic intermediate as it is suggested by the Reviewer. It can be also explain by additional thermal effects which appears in the aqueous system when additional salt is added.

  1. In mechanism, the authors suggested the photolysis step in number 2, the authors don’t write any thing in the experimental work about the control of light, is light effect was studied or no?

Reply: There was not control of light – all samples were studied at the same devices conditions. We mention about hydrolysis in the context of appearance of second form of Pt(IV). After re-examining all the results, this thread is unconfirmed.

  1. the comparison between sodium borohydride and ascorbic acid should be clear in terms of rate constants and thermodynamic parameters in addition to suggested mechanism

Reply: The case of kinetic study with sodium borohydride was cited because, during these studies, for the first time, we observed the influence of the oxygen on the reaction rates and mechanism. It was observed the rate of reaction in the case of deaerated solution is much faster comparing to the experiments carried out in standard conditions (i.e. solution contains oxygen). In this work we observe similar behavior and mechanism change. However mechanisms are quite different due to the solution composition and implications related to ascorbic acid and sodium borohydride properties.

Comparing thermodynamic values were gathered in Table 1.

Table 1. Comparison of thermodynamic parameters obtained for the Pt(IV) - NaBH4 and Pt(IV)-H2Asc system.

Parameter

Pt(IV)-NaBH4

Pt(IV) – H2Asc

Enthalpy, kJ/mol

29.6

52

Entropy, J/mol·K

−131

-72

Reviewer 3 Report

Some terms are rather unusual and seem to indicate a certain lack of general knowledge.

I’ve never heard of  a reaction being pseudo-first order.

Given the purpose of the paper, and the data collected, I think that the studies should be carried originally in the absence of oxygen and then be challenged at known oxygen concentrations.

Table 1 seems out of a lab-book not for a paper.

Reading the spectra in Figure 1 is impossible; they should be uniformed in extinction coefficients. In case specific concentration conditions are needed (I think they are), the initial and final expected spectra should be shown.

All the kinetics terms in section 3.2 is known from textbooks, no need to write it again.

I do not understand the origin of the data in Table 2. k1 should be constant, also k2 (and equal to kobs2).

Eyring and Arrhenius plots should be done on the second order rate constants.

There is in the literature a LOT of information about the speciation of hexachloridoplatitate(IV) and tetrachloridoplatinate(II). These also intervene in pH and ionic strengths, so some of the reactions in Table 8 have known parameters that could be included in the approach taken.

As a whole, the paper presents an overview of a rather complex reaction that has not tried to be simplified in the first place to gain a better understanding. Perhaps the results, as they are, might be good in some aspects, but lack a serious molecular approach as it seems the authors have intended.

Some sentences are difficult to understand.

Author Response

Review 3

Comments and Suggestions for Authors

Some terms are rather unusual and seem to indicate a certain lack of general knowledge. I’ve never heard of  a reaction being pseudo-first order.

Reply: The term “pseudo-first order” relates to experimental conditions. The character of observed kinetic curve has an exponential character. It suggest first order rate constant. But, in the case, when in the reacting system we have a huge amount of reductant, thus, its changes are smaller than that resulting from stoichiometry. Thus, we can assumed, that concentration of ascorbic acid with time do not change and it is known as pseudo conditions and rate constants is pseudo firs-order, but in fact is a second-order rate constant. The application of pseudo condition allows for easy observed rate constants determination and then second-order rate constant determination.

Given the purpose of the paper, and the data collected, I think that the studies should be carried originally in the absence of oxygen and then be challenged at known oxygen concentrations.

Reply: The Reviewer is right, but such approach means many new experiments, which will be done in the future. Because these studies will be continued.

Table 1 seems out of a lab-book not for a paper.

Reply: It is difficult to agree with Reviewer statement, because such Table were used in our previous work and in our opinion properly summarised applied conditions. However, we found one empty row and in the new version of the manuscript it was corrected.

Reading the spectra in Figure 1 is impossible; they should be uniformed in extinction coefficients. In case specific concentration conditions are needed (I think they are), the initial and final expected spectra should be shown.

Reply: We decided to left in the manuscript Fig. 1. Our intention was to show spectra coming from reagents in the full wavelength range. We calculate also extinction coefficient according to Reviewer’s suggestion. However we see also limitation of this relation (Fig. 1a, b).

(a)

(b)

(c)

(d)

Figure 1. Extinction coefficient in wavelength function obtained for: 0.05 mM (a) and 2.0 mM (b) solution of Pt(IV) and spectrum magnification in the range of wavelength 300 – 600 nm for different concentration, i.e. 0.05 mM (c) and 2.0 mM (d).

Fig. 1a shows extinction calculated for 0.05 mM solution of Pt(IV), whereas, Fig. 1b for higher concentration and spectrum magnification in the range 300 – 600 nm (Fig. 1 c vs 1d). As it can be seen at these wavelength range (Fig. 1a, c), the value of absorbance is low, thus the value of extinction coefficient varied (Fig. 1c). Comparing these data i.e. Fig. 1 c and Fig. 1 d at 453 nm we get different value of molar coefficient, i.e. 500 (0.05 mM) and 458 (2 mM).

All the kinetics terms in section 3.2 is known from textbooks, no need to write it again.

Reply: The Reviewer is right, but we believe, that kinetic description allows to readers to better understand the overall merit of the paper.

I do not understand the origin of the data in Table 2. k1 should be constant, also k2 (and equal to kobs2).

Reply: From experimental data and fitted equation we get directly observed pseudo first-order rate constants, than these values are recalculated in order to get second-order rate constant, taking into account dependency k1 = k1, obs/[H2Asc] (15), where [H2Asc] – concentration of reductant. By plotting the dependence of the rate constant (e.g. k1,obs) as a function of the initial concentration of the reductant, the values of the second - order rate constants, e.g. k1 can be determined. The linear relationships (see, Figure 4a) as well as the fact that it has its origin at (0, 0) confirms the correctness of the adopted expression for the observed pseud-first rate constant, k1,obs (15).

Eyring and Arrhenius plots should be done on the second order rate constants.

Reply: Eyring and Arrhenius plots were corrected.

There is in the literature a LOT of information about the speciation of hexachloridoplatitate(IV) and tetrachloridoplatinate(II). These also intervene in pH and ionic strengths, so some of the reactions in Table 8 have known parameters that could be included in the approach taken.

Reply: According to Reviewer’s suggestion Tables were removed, and mechanism was rewrite.

As a whole, the paper presents an overview of a rather complex reaction that has not tried to be simplified in the first place to gain a better understanding. Perhaps the results, as they are, might be good in some aspects, but lack a serious molecular approach as it seems the authors have intended

Round 2

Reviewer 2 Report

Accepted

It’s ok